# Basic Life Support Knowledge among Junior Medical and Dental Students, Communication Channels, and the COVID-19 Pandemic

**DOI:** 10.3390/medicina58081073

**Published:** 2022-08-10

**Authors:** Gaëtan Ricci, Tara Herren, Victor Taramarcaz, Nicolas Schnetzler, François Dupuis, Eduardo Schiffer, Mélanie Suppan, Laurent Suppan

**Affiliations:** 1Division of Emergency Medicine, Department of Anesthesiology, Clinical Pharmacology, Intensive Care and Emergency Medicine, University of Geneva Hospitals and Faculty of Medicine, 1211 Geneva, Switzerland; 2Ecole de Santé de Suisse Romande (ESSR), 1207 Geneva, Switzerland; 3Division of Anesthesiology, Department of Anesthesiology, Clinical Pharmacology, Intensive Care and Emergency Medicine, University of Geneva Hospitals and Faculty of Medicine, 1211 Geneva, Switzerland

**Keywords:** basic life support, undergraduate medical education, cardiopulmonary resuscitation, web-based questionnaires, COVID-19, communication

## Abstract

*Background and objective*: The prognosis of cardiac arrest victims strongly depends on the prompt provision of Basic Life Support (BLS) maneuvers. Medical students should therefore be proficient in this area, but many lack essential BLS knowledge. The goal of this prospective, closed web-based study was to determine whether a short intervention designed to motivate first-year medical students to follow a blended BLS course could lead to a significant improvement in BLS knowledge in the following year. *Materials and Methods*: A fully automated web-based questionnaire was administered to second-year medical students one year after they had been given the opportunity of following a blended BLS course (e-learning and practice session). The primary outcome was the difference, on a 6-question score assessing essential BLS knowledge, between these students and those from the 2020 promotion since the latter had not been offered the optional BLS course. *Results*: The score was similar between the two study periods (3.3 ± 0.8 in 2022 vs. 3.0 ± 1.0 in 2020, *p* = 0.114), but no firm conclusion could be drawn since participation was much lower than expected (17.9% in 2022 vs. 43.7% in 2020, *p* < 0.001). Therefore, a second questionnaire was created and administered to understand the reasons underlying this low participation. *Conclusions*: There was a lack of improvement in BLS knowledge in second-year medical students after the introduction of an optional introductory BLS course in the first-year curriculum, but the limited participation rate precludes drawing definitive conclusions. Ineffective communication appears to be the cause of this low participation rate, but a lack of motivation in the aftermath of the COVID-19 pandemic cannot be ruled out. Corrective actions should be considered to enhance communication, restore motivation, and ultimately improve BLS knowledge among medical and dental students.

## 1. Introduction

The prognosis of out-of-hospital cardiac arrest (OHCA) victims strongly depends on the presence of witnesses able and willing to perform Basic Life Support (BLS) manoeuvres [1,2,3]. Despite strong expectations among the population, and even though they may also face cardiac arrest situations during their clinical training [4], most medical students lack essential BLS knowledge [5,6].

Medical and dental students of the University of Geneva Faculty of Medicine (UGFM) are no exception. Indeed, a recent study showed that essential BLS knowledge was lacking in medical and dental students who had passed their first-year examinations but had not yet undergone UGFM resuscitation courses [7]. Taking all these elements into account, senior UGFM students postulated that training their junior colleagues at an early stage of their studies could not only enhance their knowledge but also motivate them to join a first responder network [8]. Such networks have been deployed in many different countries to decrease the time elapsed between collapse and BLS initiation [9]. The operating principle all of these networks rely on is to enable emergency medical dispatchers to alert the nearest available members in the event of an OHCA [10,11].

Therefore, in January 2021, senior UGFM students delivered a short, online, motivational intervention designed to motivate their junior colleagues to follow a blended learning course designed as a flipped classroom [12]. Briefly, a flipped classroom is a pedagogical strategy and a type of blended learning that aims to reduce time constraints, increase student engagement and improve knowledge acquisition by providing theoretical knowledge asynchronously prior to a live course. The time available for the live course can therefore be dedicated to interactive activities rather than to the transmission of theoretical knowledge [13,14]. In this particular instance, the asynchronous intervention consisted in an e-learning module, which was completed by 124 of the 529 first-year UGFM students 124 (23.4%). Of those, 97 attended the practice sessions and 48 (9.1%, 48/529) finally joined Save a Life, the first responder network operating in Geneva, Switzerland. Even though the main purpose of this learning path was to increase the number of first responders [9], it was also thought that providing first-year medical and dental students with an optional resuscitation course embedding a distance, asynchronous e-learning module could significantly improve their knowledge regarding BLS manoeuvres.

The main objective of this study was to determine whether second-year medical students who had been offered the possibility of following an optional BLS course during their first year of study had a higher BLS knowledge than those who had not been offered this possibility. This comparison was made possible since BLS knowledge data acquired in the course of a previous study in which second-year medical and dental UGFM students participated is freely available online [7,15]. The students who participated in this latter study had not been given the opportunity of following an optional BLS course during their first year of study. The second objective of the present study was to explore the reasons underlying the unexpectedly low participation rate.

## 2. Materials and Methods

### 2.1. Study Design

This was a two-tiered, prospective, closed web-based study carried out in Geneva, Switzerland, between October 2021 and May 2022. The Swiss Federal Act on Research involving Human Beings does not apply to such studies [16], but the approval of the local Ethics Committee was nevertheless sought. This Committee issued a “no objection” declaration (Req #2021-00439). The Checklist for Reporting Results of Internet E-Surveys (CHERRIES) was used to detail the methods and results [17].

### 2.2. Online Platform

Both questionnaires were created under a Joomla! 3.10 platform (Open Source Matters, New York, NY, USA) using the Community Surveys 5 component (BulaSikku Technologies Private Limited, Hyderabad, India). Answers were recorded in an encrypted MySQL-compatible database (MariaDB 5.5.5, MariaDB Foundation, Wakefield, MA, USA) hosted on a Swiss server (Kreativmedia GmBH, Zürich, Switzerland). The questionnaires were thoroughly tested prior to the start of each phase of this study. To limit attrition [18,19], branching logic was used to display relevant questions only. Completeness and regular expression (RegEx) checks were automatically performed at the end of each page to avoid missing or inconsistent data. No tracking identifier (cookies or IP address) was used since this was a closed study.

### 2.3. First Questionnaire

The first questionnaire was designed to assess BLS knowledge (Table 1).

Briefly, medical and dental UGFM students share a common pathway until the end of their second year of medical school, and represented a convenient sample of 190 potential participants. The BLS course that these students follow during their second year is based on a flipped classroom principle: all students are expected to follow an interactive e-learning module designed to provide them with theoretical knowledge regarding OHCA and BLS maneuvers prior to attending two successive practice sessions, each lasting 2 h. For the last few years, all the information about the courses and their prerequisites are exclusively hosted on a university Moodle platform, and students are not provided with any printed material. E-learning modules are also hosted on this platform. Therefore, a method similar to that outlined by Sturny et al. was used to recruit second-year medical and dental students [7], i.e., a slide inviting them to answer the questionnaire was added at the very beginning of the e-learning module. On 18 February 2022 (ten days before the first practice session), the faculty secretariat sent an email to the whole promotion reminding them that the e-learning module had to be completed prior to attending the training sessions.

All participants were informed that answering this questionnaire was optional and that they could choose to withdraw at any time. A data policy statement was displayed on the main page, and the email address of the main investigator was displayed. Consent was gathered electronically. Since the participants were never asked to enter their identity or email address, the answers recorded were irreversibly anonymized.

### 2.4. Second Questionnaire

Since participation was much lower than expected (43.7%, 80/183 in Sturny’s study vs. 17.9%, 34/190, *p* < 0.001), a second questionnaire was designed to determine the reasons underlying this low participation rate (Table 2). The link to this questionnaire was broadcasted on the WhatsApp group of the second-year promotion on 23 May 2022. A single reminder was sent using the same communication channel two days later.

### 2.5. Exclusion Criteria

For the first questionnaire, incomplete responses, responses recorded by BLS instructors, by certified health care professionals or by participants who answered that they were not registered as medical students were excluded

Regarding the second questionnaire, only incomplete responses were excluded from analysis. There were no other exclusion criteria for this questionnaire.

### 2.6. Outcomes

The primary outcome was the difference on the 6-point “essential BLS knowledge” score, between the medical students who had participated in Sturny’s study in 2020 [7] (“medical students 2020” group) and those who participated in the current study (“medical students 2022” group). Conversely to students belonging to the “medical students 2022” group, those belonging to the “medical students 2020” group had not been offered the possibility of following a first aid course during their first year. The same difference on the 10-point score was also analysed. Responses to the second questionnaire were used to determine the proportion of students who answered that they had fully or partially followed the e-learning module. The reasons for not following the e-learning module or for following it only partially were also analysed.

### 2.7. Data Curation and Statistical Analysis

Stata 17.0 (StataCorp LLC, College Station, TX, USA) was used for data curation and statistical analysis. To avoid any assessor-linked bias, the original Stata DTA file from Sturny’s study was retrieved [15] and free text answers were re-coded by one investigator (GR) and reviewed by a second investigator (LS). The same procedure was applied to free text answers gathered in the course of the current study (Table 1). Disagreements were resolved by consensus.

Normality was assessed graphically. In case of doubt, the Kolmogorov-Smirnov test was used. For binary and categorical variables, between-group comparisons were carried out using Fisher’s exact test or the chi-squared test depending on cell size (cut-off: 5) and normality. For the 6-point “essential BLS knowledge” score and for the 10-point score, each correct answer was worth 1 point with no differential weighting. Once computed, scores were treated as continuous variables and between-group comparisons performed using Student’s *t*-test. These scores are presented as means ± SD (standard deviation). A *p* value < 0.05 was considered significant.

## 3. Results

### 3.1. First Questionnaire

Out of the 190 second-year medical and dental students registered at UGFM, 34 (17.9%) provided responses that were finally analyzed (Figure 1). Participation was not statistically different between medical and dental students (19.6%, 32/163 in medical students vs. 7.4%, 2/27 in dental students, *p* = 0.176).

Authors should discuss the results and how they can be interpreted from the perspective of previous studies and of the working hypotheses. The findings and their implications should be discussed in the broadest context possible. Future research directions may also be highlighted

The age of the participants was similar (22 ± 4 years old in 2020 vs. 21 ± 2, *p* = 0.08). Gender distribution was also similar (*p* = 0.26).

There was no statistically significant difference on the 6-point essential BLS knowledge score between the 2020 and the 2022 promotions (3.0 ± 1.0 in 2020 vs. 3.3 ± 0.8, *p* = 0.114). The 10-point score was also similar between these groups (5.9 ± 1.7 in 2020 vs. 6.0 ± 1.6, *p* = 0.748).

### 3.2. Second Questionnaire

A total of 38 students started the second questionnaire and 34 completed it. The majority answered that they had followed the e-learning prior to attending the practice sessions (19/34, 56%). Most of them answered that they had fully completed this module (17/19, 89.5%), with only two of them answering that they had followed it only in part (10.5%). Of those, one answered that they had forgotten to resume it, while the other reported a lack of motivation.

More than a third of all responders (12/34, 35.3%) answered that they had not followed the e-learning module. Only two students reported a lack of motivation (2/12, 16.7%), while all others answered that they did not know that this e-learning module existed (10/12, 83.3%). These latter students unanimously answered that they had not seen a link to the e-learning module on the Moodle platform and that they had not received (3/10, 30%) or did not remember receiving (7/10, 70%) an e-mail informing them that this module had to be followed prior to attending the practice sessions.

## 4. Discussion

### 4.1. Main Considerations

This study shows a lack of significant improvement in essential BLS knowledge among second-year medical students after offering them an optional BLS training course during their first year of study. Among the hypotheses which could help explain this finding, the low participation rate must be acknowledged first. Indeed, while Sturny et al. reported a 44% participation rate in 2020 [7], participation was more than twice as low (17.9%) in this study. Therefore, the present study certainly lacks power on the one hand, while selection bias is quite likely on the other.

The reasons behind this low participation rate are not easy to determine, but several hypotheses can be drawn. Even though the COVID-19 pandemic and its never-ending waves have markedly altered the motivation of health care students [20], lack of motivation was seldom given as a reason for not following the e-learning module in the present study. Of course, this could be linked to a selection bias, and those who lacked the motivation to follow the e-learning module or to answer the first questionnaire would also be unlikely to answer the second questionnaire.

Many students reported that they did not know that this e-learning module existed. Conversely to those who participated in Sturny’s study [7], the medical and dental students who were part of the 2022 promotion did not receive a printed booklet which included information regarding the courses and workshops they had to attend and were expected to find all this information on the Moodle platform. In addition, even though an email reminding them that the e-learning module had to be followed prior to attending the BLS practice sessions was sent to the whole promotion by the UGFM secretary, many students still answered that they had not received such an email or that they did not remember receiving it. These elements prompt several considerations. First, and even though digitization is desirable on many accounts [21], a particular emphasis should be put on visualization modalities [22] and on accessibility [23]. Indeed, quickly drawing a printed booklet from one’s backpack to check a schedule or course prerequisites is rather straightforward, while browsing through a learning management platform can prove both tedious and difficult [24,25]. In addition, the mail epidemic many certified health care professionals have to deal with on a daily basis [26,27] seems to be extending to medical students, many of whom may not be able to take in the flood of incoming information [28].

The changes brought in the medical education curriculum by the COVID-19 pandemic may also have influenced the participation rate negatively. Indeed, the vast majority of the 190 medical and dental students who could have participated in this study began their studies during the pandemic. Therefore, instead of attending live classes, they mainly followed an asynchronous “courses-on-demand” curriculum [29]. Therefore, these students were not required to follow their courses in any particular order. Furthermore, many adapted their curriculum to their personal schedule. All these factors could have participated in preventing the members of this particular promotion from anticipating the presence of potential prerequisites which had to be followed prior to attending certain sessions.

### 4.2. Limitations

This study has several limitations, the most important of which have already been acknowledged in the previous sections. It should also be acknowledged that medical students who had repeated their second year and who had therefore been unable to participate in the optional course offered to first-year students were not excluded. The risk of bias linked to this limitation is however rather low since attrition is highest at the end of the first year of medical school and the number of repeaters is very low during the following years.

### 4.3. Perspectives

Several improvements can be considered in the immediate aftermath of this study. Among these, the most important is the assessment of current communication channels used to spread information among medical and dental students. This assessment should help design more appropriate tools to ensure effective communication and avoid information loss. In addition, the unforeseeable consequences of the COVID-19 pandemic on the way medical and dental students consider their studies deserve further assessment and corrective actions could be designed accordingly.

## 5. Conclusions

This study shows a lack of improvement in BLS knowledge in second-year medical students after the introduction of an optional introductory BLS course in the first-year curriculum. This result should however be put into perspective because of the low participation rate. This low rate seems to be related to communication issues and, probably, to curriculum changes generated by the COVID-19 pandemic, as well as to a lack of motivation. Corrective actions should now be considered and taken to enhance communication, restore motivation, and ultimately improve BLS knowledge among medical and dental students.

## Figures and Tables

**Figure 1 medicina-58-01073-f001:**
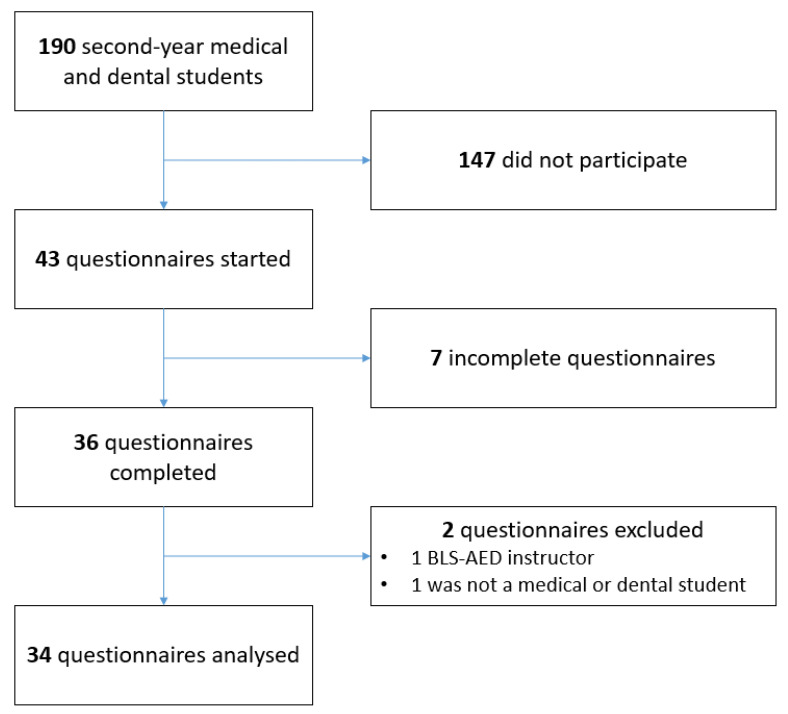
Flowchart of the inclusion of questionnaires completed by medical and dental students.

**Table 1 medicina-58-01073-t001:** First questionnaire.

Survey Page, Field, and Question	Type of Question
1—Demographics	
How old are you?	RegEx
What gender do you identify with?	MCQ
2—Prior training	
Have you ever taken a first aid course?	Yes/No
What first aid training have you already received? ^1^	MAQ ^2^
For what reason(s) did you not attend a first aid course before? ^1^	MAQ ^2^
Do you or have you engaged in any of these activities?	MAQ
Are you a graduate of a health profession?	Yes/No
You are (paramedic/nurse/doctor): ^1^	MAQ ^2^
Would you like more first aid training?	MCQ
3—First responders ^3^	
Did you become a first responder?	Yes/No
Are you still active as a first responder? ^1^	Yes/No
4—Student in the health sector	
Are you a student of a health profession?	Yes/No
In which field? ^1^	MAQ ^2^
What field of medicine are you studying? ^1^	MAQ ^2^
Do you plan on specializing in acute medicine (intensive care, anesthesiology, emergency) ^1^	MCQ
Did you undertake the BLS-AED e-learning course offered to first-year medical students? ^1^	MCQ
Did you also attend the practice session? ^1^	Yes/No
5—BLS knowledge I	
Have you ever heard of BLS/ACLS?	Yes/No
What does AED stand for? ^4,6^	Free text
In what year were the first aid guidelines last revised?	RegEx
Which number should be called in the event of a medical emergency? ^4,5,7^	Free text
6—BLS knowledge II	
Which criteria should be met to identify cardiac arrest? ^4,5^	MAQ
In what order should you proceed to treat a patient in cardiac arrest? ^4,5^	Ordering
What is the most suitable artery to detect a pulse in an adult patient? ^4^	MAQ
How deep should chest compressions be performed on an adult? ^4,5^	MCQ
Which compression to ventilation ratio should be used? ^4^	MCQ
What is the recommended rate of chest compressions? ^4,5^	MCQ
Are chest compressions useful if there is no ventilation? ^4,5^	MCQ
What is the first recommended action for an adult patient who is choking, given that they are unable to speak or cough? ^4^	MCQ
On a scale of 1 to 10, how comfortable would you say you would be in a resuscitation situation?	1–10 scale

BLS: basic life support; MAQ: multiple answer question (one or more answers); MCQ: multiple choice question (only one answer); RegEx: regular expression (validation rule). ^1^: branching logic was used to display this question, ^2^: free answers could be recorded,^3^: this page was only displayed to participants who answered that they had followed a first aid course to become first responder. ^4^: question used to compute the 10-point BLS score, ^5^: question used to compute the 6-point “essential BLS knowledge” score, ^6^: all answers including the word “defibrillator”, regardless of its spelling, were accepted as correct, ^7^: answers were counted as correct if either 112, 144 or 911 were entered by the participant.

**Table 2 medicina-58-01073-t002:** Second questionnaire.

Survey Page, Field, and Question	Type of Question
Page 1	
Did you follow the e-learning on the Moodle platform before attending the first resuscitation practice session?	MCQ
Why (several answers possible)? ^1^	MAQ
Did you see a link to e-learning module on Moodle? ^1^	Yes/no
Did you see an invitation to answer a master thesis questionnaire? ^2^	MCQ
Why did you only partially follow the e-learning (several answers possible)? ^3^	MAQ
Page 2	
Did you receive an information e-mail advising you to follow this e-learning course?	MCQ
Page 3	
Did you follow the resuscitation course offered in the first year?	MCQ

MAQ: multiple answer question (one or more answers); MCQ: multiple choice question (only one answer). ^1^: question displayed only to those who answered that they did not follow the e-learning module, ^2^: question displayed only to those who answered that they had followed the e-learning module (entirely or in part), ^3^: question displayed only to those who answered that they had followed the e-learning module only partially were accepted as correct.

## Data Availability

The curated data file is available on Mendeley Data: https://doi.org/10.17632/52yy8jm3by.1.

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
