# Peer review of "Basic Life Support Knowledge among Junior Medical and Dental Students, Communication Channels, and the COVID-19 Pandemic"

_medicina, 2022, doi:10.3390/medicina58081073_

Round 1

Reviewer 1 Report

The study topic is interesting and it will certainly be useful for the readers.

The manuscript is scientifically written well.

All the sections are appropriately described.

The scope of the study is crisply highlighted.

Methodology section is logically narrated.

Statistical tests used for analysis are appropriate.

Results are aptly explained, tables and figures are good for presentation of the data.

Discussion of the findings from the study are suitably compared and discussed with other similar studies.

Relevant citations are provided wherever it was necessary.

References are appropriate including the recent studies.

Reviewer 2 Report

Overall the content of the manuscript is good with an interesting research theme. It should be added to the discussion section regarding the comparison of results between the two groups of medical and dental students and the extent of knowledge between these two groups regarding BLS.

Reviewer 3 Report

Sending e-mails to students is not enough because some e-mails can be found in a SPAM box. I agree that a communication problem was one of the reason of low response.

Reviewer 4 Report

The research showed communication problems that appeared in the pandemic year.Despite the numerous tools provided by the universities, students did not show enthusiasm for reviewing all the sent materials, and it should be investigated whether there is a reduced interest in other topics as well. I think that for both medical and dental students, teaching should take place in situ, where it can be seen that the role of the lecturer as a motivator is very important.